# Expression of a PCSK9 Gain-of-Function Mutation in C57BL/6J Mice to Facilitate Angiotensin II-Induced AAAs

**DOI:** 10.3390/biom12070915

**Published:** 2022-06-29

**Authors:** Hisashi Sawada, Alan Daugherty, Hong S. Lu

**Affiliations:** 1Saha Cardiovascular Research Center, College of Medicine, University of Kentucky, Lexington, KY 40536, USA; alan.daugherty@uky.edu (A.D.); hong.lu@uky.edu (H.S.L.); 2Saha Aortic Center, College of Medicine, University of Kentucky, Lexington, KY 40536, USA; 3Department of Physiology, College of Medicine, University of Kentucky, Lexington, KY 40536, USA

**Keywords:** PCSK9, hypercholesterolemia, adeno-associated virus, angiotensin, aortic aneurysm, mouse

## Abstract

Angiotensin II (AngII) infusion in mice has been used widely to investigate mechanisms of abdominal aortic aneurysms (AAAs). To achieve a high incidence of AngII-induced AAAs, mice should be hypercholesterolemic. Therefore, either low-density lipoprotein receptor (LDLR) or apolipoprotein E deficiency have been used as a hypercholesterolemic background. However, it is a time-consuming and expensive process to generate compound deficient strains that have either an LDLR or apolipoprotein E deficient background. Proprotein convertase subtilisin/kexin type 9 (PCSK9) facilitates the degradation of LDL receptors. Previous studies demonstrated profound increases of plasma cholesterol concentrations after a single intraperitoneal injection of adeno-associated viruses (AAV) expressing a gain-of-function mutation of mouse PCSK9 (AAV.mPCSK9^D377Y^) in C57BL/6J mice fed a Western diet. Of note, injection of AAV.mPCSK9^D377Y^ augmented AngII-induced AAA formation in C57BL/6J mice that had comparable severity of AAAs to LDLR deficient mice. Thus, AAV.mPCSK9^D377Y^ infection greatly expedites studies on a gene of interest using AngII-induced AAAs. This commentary provides a brief technical guide of this approach and discusses the pros and cons of its use in AAA research.

## 1. Introduction

Abdominal aortic aneurysms (AAAs) have a devastating impact on public health, but there are no validated medications for preventing the initiation, progression, or rupture of AAAs. Continued mechanistic explorations are needed to establish new therapeutics. Subcutaneous infusion of angiotensin II (AngII) leads to aortic aneurysm formation in the thoracic and suprarenal abdominal regions of mice [1,2]. Many studies have used this mouse model to investigate the mechanism of aortic aneurysms in both regions [3,4,5]. An important aspect of this mouse model is hypercholesteremia to augment incidence of AAAs [6,7]. The incidence of AAAs in AngII-infused mice is less than 20% in a normocholesterolemic background, whereas it is generally increased more than 80% in hypercholesterolemic mice [4]. Low-density lipoprotein receptor (LDLR) or apolipoprotein E deficient mice have been used widely as the hypercholesterolemic backgrounds to enhance AngII-induced AAAs.

Genetic manipulation in mice is a common and optimal approach for determining molecular mechanisms of AAAs. However, it is a time- and cost-consuming process to breed a gene of interest into a hypercholesterolemic background. At least 3 breeding cycles and approximately 2 years are needed to generate a sufficient number of mice that have compound genetic manipulations of a specific gene and either LDLR or apolipoprotein E [8]. Considerable costs are also needed for the management of mouse colonies, including housing and genotyping. Therefore, the development of a mouse colony with genetic manipulations in a hypercholesterolemic background is a significant barrier in AAA research in which the pathology is generated by chronic infusion of AngII.

Proprotein convertase subtilisin kexin 9 (PCSK9), a member of the subtilisin serine protease family, exerts an important role in cholesterol metabolism [9]. PCSK9 binds to LDLRs and facilitates its degradation. A gain-of-function mutation D374Y in PCSK9 causes autosomal-dominant hypercholesterolemia in humans [10]. Subsequently, preclinical studies demonstrated increased plasma cholesterol concentrations after a single injection of adeno-associated viral vector (AAV) expressing PCSK9 with either the human D374Y or mouse D377Y (the human equivalence of D374Y) mutants in mice [11,12]. This mouse model mimics the hypercholesterolemic phenotype of LDLR deficient mice in exhibiting high plasma non-high-density lipoproteins concentrations and accelerated atherosclerosis formation when fed a Western diet [11,12,13]. Of note, injection of AAVs containing the mouse PCSK9^D377Y^ (AAV.mPCSK9^D377Y^) also augmented AngII-induced AAA formation in C57BL/6J mice, with comparable AAA severity to LDLR deficient mice with AngII infusion [14,15,16,17]. Thus, injection of AAVs containing a PCSK9 gain-of-function mutant saves the cost and time for generating mice to an LDLR deficient background. In addition, this mode can induce hypercholesterolemia in both young and adult mice [18]. Therefore, injection of AAVs containing a PCSK9 gain-of-function mutant is an easy and readily used approach to augment AngII-induced AAAs in C57BL/6J mice. In this commentary, we introduce the injection of AAV.mPCSK9^D377Y^ to induce hypercholesterolemia in C57BL/6J and provide suggestions for its optimal use to facilitate AngII-induced AAAs.

## 2. Brief Protocol for AAV.mPCSK9^D377Y^ Injection

The following descriptions are experimental steps for AAV.mPCSK9^D377Y^ injection into mice [18]. Please note that injection of AAV is biosafety level 1.
Calculate required AAV.mPCSK9^D377Y^. A total of 10–20 × 10^10^ genomic copies are recommended per mouse. It is recommended to calculate AAVs based on genomic copies, not body weight.Dilute AAVs with cold sterile phosphate-buffer saline.Aliquot 200 µL (10–20 × 10^10^ genomic copies) of AAV dilation into 1.5 mL sterile tubes.Draw the aliquoted AAV dilution into an insulin syringe (31G).Inject AAVs intraperitoneally into mice.

Note: Change the food to a Western diet immediately after injection. Osmotic pumps for AngII infusion should be implanted 1–2 weeks after injection of AAVs. Continue Western diet during AngII infusion.

## 3. Representative Results

Plasma PCSK9 concentrations are increased in response to AAV.mPCSK9^D377Y^ injection (Figure 1A). Plasma cholesterol concentrations are increased by feeding a Western diet in both AAV.mPCSK9^D377Y^ injected and LDLR deficient mice (Figure 1B). Of note, cholesterol concentrations in mice with AAV.mPCSK9^D377Y^ injection are lower than that of LDLR deficient mice. AngII infusion leads to aortic aneurysm formation in the suprarenal abdominal aorta of both AAV.mPCSK9^D377Y^ injected and LDLR deficient mice (Figure 2B), and AngII-induced aortic dilatations are comparable between the two mouse strains (Figure 2B). Aortic ruptures are observed in 10 to 30% of AngII-infused mice infected with AAV.mPCSK9^D377Y^ [15]. The incidence of aortic ruptures is comparable between C57BL/6J mice infected with AAV.mPCSK9^D377Y^ and LDLR deficient mice [15,19]. AAAs in AAV.mPCSK9^D377Y^-infected mice display macrophage accumulation and increases in mRNA abundance of inflammatory cytokines, such as *Il1b* and *Tnfa* [16].

## 4. Expense for AAV.mPCSK9^D377Y^

It is worth noting that hypercholesterolemia can be established in adult mice by a single injection of AAV.mPCSK9^D377Y^. In addition, AAVs can be generated at low costs (<$10/mouse). AngII-induced AAAs in mice with AAV.mPCSK9^D377Y^ injection exhibit comparable severity to those of LDLR deficient mice. Therefore, AAV.mPCSK9^D377Y^ injection is an optimal and cost-effective approach for augmenting AngII-induced AAAs in C57BL/6J mice.

## 5. Impact of AAV.mPCSK9^D377Y^ on Plasma Total Cholesterol Concentrations

Despite comparable phenotypes of AngII-induced AAAs, plasma total cholesterol concentrations are lower and have more variations in mice with AAV.mPCSK9^D377Y^ than in LDLR deficient mice [18]. Shortly after the initiation of Western diet feeding, both models display significant increase of total cholesterol concentrations. On the basis of our long-term experience, plasma total cholesterol concentration could range from 500–1500 mg/dl between 2–6 weeks of Western diet feeding in mice injected with AAV.mPCSK9^D377Y^.

Figure 1B shows an example of plasma cholesterol concentrations after AAV.mPCSK9^D377Y^ injection in our pilot study [18]. Despite comparable incidence and severity of AngII-AAAs, plasma cholesterol concentrations are different slightly between AAV.mPCSK9^D377Y^-infected and LDLR deficient mice. Cholesterol concentrations are greater than 1000 mg/dL at 6 weeks of Western diet in both two models. However, plasma cholesterol concentrations are approximately 400 mg/dL lower in AAV.mPCSK9^D377Y^ infected mice compared to LDLR deficient mice. As demonstrated in our previous study [7,20], relatively modest hypercholesterolemia augments AngII-induced AAAs; however, unlike atherosclerosis, AngII-induced AAAs are not further augmented with further increases of plasma total cholesterol concentrations. Since plasma cholesterol concentration is an important determinant in atherosclerosis formation, the difference of cholesterol concentrations should be considered when studying atherosclerosis using this model. A previous study using AAVs containing human PCSK9^D374Y^ mutant showed sex dimorphic effects on plasma PCSK9 concentrations and atherosclerosis formation in LDLR deficient mice, which should also be taken in a consideration [21]. Although there are several caveats for its use in atherosclerosis study, infection with AAV.mPCSK9^D377Y^ is a rigorous mode for the augmentation of AngII-induced AAAs in mice.

## 6. Non-Responders to AAV.mPCSK9^D377Y^ Injection

It is frequently observed that ~10% of mice injected with AAV.mPCSK9^D377Y^ have no profound increases of plasma PCSK9 or cholesterol concentrations. Therefore, it is recommended to have prospective exclusion criteria to remove non-responders. For instance, we remove mice from the study if their plasma total cholesterol concentrations are <400 mg/dL at 2 weeks and <600 mg/dL at 4 weeks of Western diet feeding. For rigor and reproducibility, it is important to include appropriate negative controls. Mice infected with null-AAV fed a Western diet and C57BL/6J fed a normal laboratory rodent diet can be used as appropriate controls.

## 7. Extended Effects of PCSK9

PCSK9 is involved in vascular inflammation independent of facilitating degradation of LDLR [22]. Thus, PCSK9 may contribute to AAA formation through not only the induction of hypercholesterolemia but also regulation of vascular inflammation. It would be interesting to investigate extended effects of PCSK9 independent of the regulation of LDLR.

## 8. Combination of AAV.mPCSK9^D377Y^ with the CreER^T2^-loxP System

The CreER^T2^-loxP system has been used widely for cell type-specific and temporally controlled genetic manipulation in mice. Importantly, this system needs a caution approach in the combination with AAV.mPCSK9^D377Y^ infection. To activate CreER^T2^, tamoxifen is most commonly administered using intraperitoneal injection after its dissolution in corn oil [23]. However, injected corn oil resides in the peritoneal cavity in a prolonged period, which may interrupt the absorption of AAV.mPCSK9^D377Y^. Given this potential issue, it is not recommended to inject AAV.mPCSK9^D377Y^ intraperitoneally after tamoxifen injections. We recommend injecting PCSK9-AAVs prior to DNA recombination by tamoxifen injection in the CreER^T2^-loxP system. Alternatively, if tamoxifen must be injected prior to AAV.mPCSK9^D377Y^ injection, intravenous injection of the AAVs is recommended.

## 9. Conclusions

This commentary provides a practical guide for the induction of a PCSK9 gain-of-function mutation in mice. AAV.mPCSK9^D377Y^ infection is a time- and cost-effective approach for augmenting AngII-induced AAA formation in C57BL/6J mice. We hope this protocol will promote preclinical studies using the AngII infusion mouse model to provide insight in mechanisms of AAAs to facilitate acceleration of the development of new therapeutics.

## Figures and Tables

**Figure 1 biomolecules-12-00915-f001:**
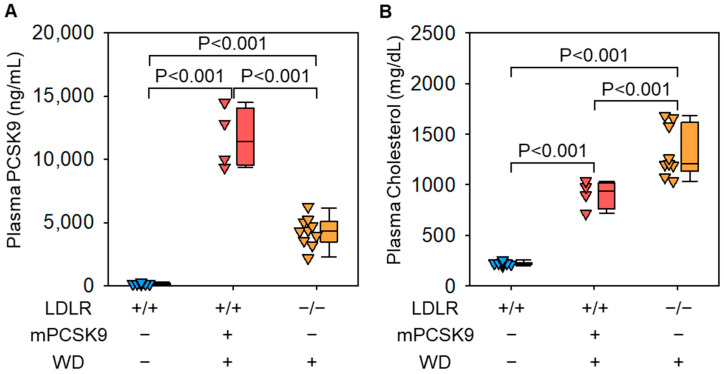
Plasma (**A**) PCSK9 and (**B**) total cholesterol concentrations in male control (C57BL/6J mice infected with AAVs containing an empty vector), AAV.mPCSK9^D377Y^ infected, and LDLR deficient mice (*n* = 4–10 per group, 10-week-old). Plasma PCSK9 and total cholesterol concentrations were measured using a mouse PCSK9 ELISA kit (MPC900, R&D) and a cholesterol E enzymatic assay kit (#439-17501, Wako), respectively. WD indicates Western type diet (TD.88137, Envigo). Data were Log10 transformed, and *p* values were determined by one-way ANOVA followed by Holm–Sidak test.

**Figure 2 biomolecules-12-00915-f002:**
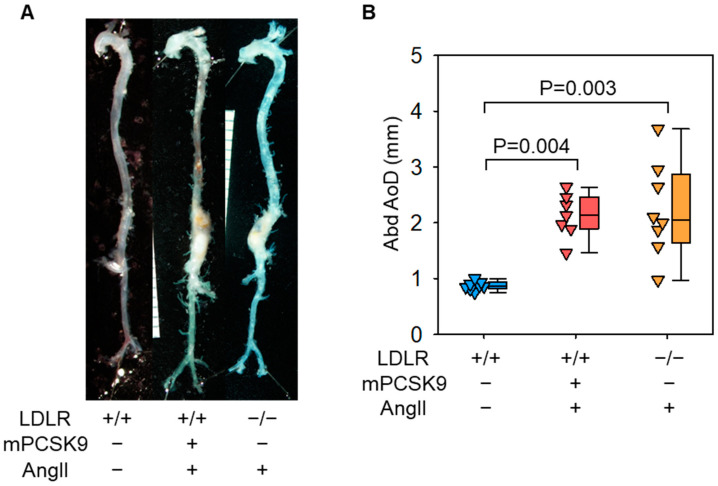
(**A**) Representative ex vivo images of the aorta and (**B**) maximal abdominal aortic diameters (Abd AoD) from male control (C57BL/6J mice infected with AAVs containing an empty vector), AAV.mPCSK9^D377Y^ infected, and LDLR deficient mice (*n* = 8–9 per group, 10-week-old). AngII (1000 ng/kg/min, H1705, Bachem) was infused through a subcutaneously implanted osmotic pump (Alzet 2004, Durect) for 4 weeks, as described previously [14]. *p* values were determined by Kruskal–Wallis followed by Tukey test.

## Data Availability

The raw data that support the representative results reported in this manuscript are available from the authors upon reasonable request.

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
