# Peer review of "Expression of a PCSK9 Gain-of-Function Mutation in C57BL/6J Mice to Facilitate Angiotensin II-Induced AAAs"

_biomolecules, 2022, doi:10.3390/biom12070915_

Round 1

Reviewer 1 Report

This is an excellent and timely commentary on a fast-moving research topic in vascular disease. Specifically, the authors provide a detailed technical guide for using a single intraperitoneal injection of adeno-associated viruses (AAV) expressing a gain-of-function mutation of mouse PCSK9 (mPCSK9-AAV) in C57BL/6J mice as a time-saving as well as cost-effective alternative for ApoE-/LDLR-deficient mice. Importantly, the authors also discussed the pros and cons of this novel approach when used it in atherosclerosis and AAA studies.  

Author Response

We appreciate the kind comments of the reviewer 1.

Reviewer 2 Report

The manuscript has nicely summarized a new model to mimic LDLR KO mice developed AAA induced by Ang II, which is very practical and interesting. Just a few minor questions:

1. How about the AAA rupture percentage and survival rate in this PCSK9-AAV injection model, is it also comparable to LDLR deficient mice?

2. Why there is a quite big variation in the aortic diameter in LDLR KO mice in figure 2B? The largest reached almost 4mm, didn't the mouse get AAA rupture or dissection?

3. How about the inflammatory markers in this new model mice if the cholesterol level is overall lower than LDLR Ko mice on a high-fat diet? 

Reviewer 3 Report

The Authros presented a well-written manuscript that provides a better way to obtain hypercholestherolemic mice in order to study pathophysiological mechamims of aortic aneurysm formation.

- the Authors should better explain why AngII-induced aneurysm are a located predonminatly in abdominal aorta instead of thoracic aorta.

- the Authors wrote that there is a signficant difference in cholesterol level between mice with LDLR deficiency and those with PCSK9 gain of function (as shown in figure 1 B). The Authors should better clarify if a relationship between Cholestherol level and AAA size exist.

- Whih therapeutic implication could have this new method?

- Which could be the time and the money saved with this procedure?
